# Explainable Convolutional Neural Network to Investigate Age-Related Changes in Multi-Order Functional Connectivity



**Sunghee Dong [1], Yan Jin [2], SuJin Bak [1], Bumchul Yoon [2,3] and Jichai Jeong [1,*]**

1  Department of Brain and Cognitive Engineering, Korea University, Seoul 02841, Korea; zpfzpf123@korea.ac.kr (S.D.); soojin7897@korea.ac.kr (S.B.)
2  Department of Physical Therapy, College of Health Science, Korea University, Seoul 02841, Korea; jinyan@korea.ac.kr (Y.J.); yoonbc@korea.ac.kr (B.Y.)
3  Major in Rehabilitation Science, Graduate School, Korea University, Seoul 02841, Korea
*  Correspondence: jcj@korea.ac.kr

**Abstract:** Functional connectivity (FC) is a potential candidate that can increase the performance of brain-computer interfaces (BCIs) in the elderly because of its compensatory role in neural circuits. However, it is difficult to decode FC by the current machine learning techniques because of a lack of physiological understanding. To investigate the suitability of FC in BCIs for the elderly, we propose the decoding of lower- and higher-order FC using a convolutional neural network (CNN) in six cognitive-motor tasks. The layer-wise relevance propagation (LRP) method describes how age-related changes in FCs impact BCI applications for the elderly compared to younger adults. A total of 17 young adults ($24.5 \pm 2.7$ years) and 12 older ($72.5 \pm 3.2$ years) adults were recruited to perform tasks related to hand-force control with or without mental calculation. The CNN yielded a six-class classification accuracy of 75.3% in the elderly, exceeding the 70.7% accuracy for the younger adults. In the elderly, the proposed method increased the classification accuracy by 88.3% compared to the filter-bank common spatial pattern. The LRP results revealed that both lower- and higher-order FCs were dominantly overactivated in the prefrontal lobe, depending on the task type. These findings suggest a promising application of multi-order FC with deep learning on BCI systems for the elderly.

**Keywords:** brain-computer interface (BCI); convolutional neural network (CNN); electroencephalogram (EEG); explainable artificial intelligence (XAI)

## 1. Introduction

With advancements in science and medical technologies, the average life span of humans has gradually increased [1]. Therefore, there is a growing need for brain-computer interface (BCI) systems for healthy elderly people going through nonpathological physical and cognitive declines [2,3]. BCI systems connect the brain to a computer, allowing the user to enhance their life [4,5]. As machine learning and intelligent robotic technology advance, the range of BCI applications is growing. The development of a hybrid BCI, such as the simultaneous use of near-infrared spectroscopy [6] and an electroencephalogram (EEG) system, further increases the potential of BCI applications to real-life situations [7–9]. Recently, the use of BCI systems to enhance health care [10] and to ensure the comfortable living of the elderly has caught the attention of many people [11]. Recent studies have found that BCIs can be a useful supplement for older people to mitigate their physical [12], cognitive [13], and mental health declines [14].

However, the applications of BCIs for the elderly are highly limited compared with those for younger adults because of the unrevealed effects of aging on the functional measures of the brain [15]. The neuronal population continues to change even after the brain is fully developed, leading to distinctive changes in brain functions throughout the life span of an individual. There is evidence for an age-related decrease in executive functions [16], processing speed [17], and the inhibition of unnecessary cognitive processes [18].

Most of the age-related functional losses are comprehensible as a result of reduced brain activity caused by structural changes such as atrophy or dedifferentiation [19]. Aging leads to shrinkage of the brain due to the decrease in gray and white matter volume [20]. This volumetric change is especially prominent in the frontal cortex [19]. The aging brain sometimes recruits brain regions in a nonselective manner, an indication of age-related dedifferentiation [21], which causes disinhibition and attentional deficiency by reducing the selectivity and specificity of neural systems [22]. These negative changes in the brain decrease classification performance in BCI research for the elderly more significantly than that of their younger counterparts [23].

However, recent findings indicate that the aging brain sometimes works harder than a younger brain. Age-related overactivation is found in various tasks and brain regions as a compensation mechanism for the neurocognitive decline of the neural system [24]. In many cases, it is accompanied by similar or better performances in the elderly compared with their younger counterparts [25,26]. Even when the elderly exhibit poorer cognitive performances than young adults, age-related overactivation is positively correlated with behavioral performance in the elderly. These results are evidence of the fact that the aging brain recruits additional brain resources to complement neurophysiological decline [24].

In particular, because of the reduced neural resources in the aging brain, the network of neuronal populations increases to process the reduced neural information more efficiently. The compensation-related utilization of neural circuits hypothesis (CRUNCH) suggests that neural circuits need to be activated further to compensate for the processing inefficiency of the aging brain by recruiting more neural resources [27]. Functional connectivity (FC) shows overactivation in the aging brain during task performance [28,29]. The compensatory increase in FC differs from dedifferentiation in that the former is beneficial for neural decline, and the latter is associated with degradation from the optimal state of neurological specialization [30,31]. The functional network at the frontal area especially exhibits greater compensatory overactivation than any other brain region to, presumably, compensate for structural atrophy [32]. The critical role of FC in the aging brain suggests that FC can be an optimal feature in BCIs for the elderly.

However, FC association with a task is not a popular feature in BCI applications because it is high-dimensional, and the way it is affected by various conditions such as age, task, and stimulation is not yet fully understood. To classify FC, previous studies attempted to extract features for a linear classifier [33], or to use network measures that characterize the properties of a connectivity map, such as the clustering coefficient [34]. Several studies on passive BCIs tried to use FC for emotion recognition [35,36] and alertness detection [37], with various network parameters such as global efficiency, degree of connectivity, and characteristic path lengths. However, feature extraction requires prior knowledge of the input data and often removes relevant information [38]. The use of network measures based on graph theory is dependent on the choice of measures, causes loss of information, and is hard to interpret intuitively.

The deep learning technique can be used to decode connectivity maps because it has been introduced in neuroscience to solve the individuality, diversity, and unpredictability of brain signals [39]. A convolutional neural network (CNN) is one of the most effective deep learning techniques for conducting automatic feature extraction and classification for high-dimensional input data. It can decode an individual's FC pattern without manual feature extraction by processing the network input, regardless of the subject's age. CNN has recently been used for EEG decoding [40–42], but it has mostly been focused on younger adults based on the temporal, spatial, or spectral information of neural activity.

In this study, we propose to decode FC in a six-class classification using CNN for the elderly and young adults. The primary reason for using FC for BCIs in this study, rather than using the more commonly used brain features, is that a neural circuit shows compensatory overactivation in the aging brain. The proposed method benefits from the compensatory role of FC, which provides advantages for the implementation of BCIs with the elderly relative to previous methods. To verify the hypothesis, low- and high-order

FCs were measured to obtain functional information in different levels of neural circuits. Multi-order FCs were estimated from five frequency bands of EEG signals during six cognitive-motor tasks. There are two main purposes for this study. First, to compare the classification performances obtained by the proposed method in the elderly to those of the younger adults. Second, to analyze the effects of age-related changes in multi-order FCs on classification performance in younger and elderly groups using layer-wise relevance propagation (LRP). This study suggests an FC-based BCI system using an explainable deep learning technique, which is particularly advantageous in research regarding aging.

## 2. Materials and Methods

### 2.1. Participants

We recruited 12 elderly people (mean age: 72.5 ± 3.2; 2 males and 10 females) and 17 young adults (mean age: 24.5 ± 2.7; 9 males and 8 females). Both groups consisted of only right-handed people. No subjects had any history of neurological or pathological diseases and had normal or corrected normal vision. The Korean Mini-Mental State Examination (K-MMSE) was conducted for the elderly to verify whether they had normal cognitive abilities [43]. The elderly with a K-MMSE score of 24 or above were allowed to participate in this experiment. Therefore, all the older participants were determined to be experiencing nonpathological normal aging. All of the subjects gave written informed consent, which was approved by the Korea University Institutional Review Board (NO. 17-126-A-2).

### 2.2. Apparatus

We used a research-grade wireless dry EEG device (DSI-24, WEARABLE Sensing, San Diego, CA, USA) for data acquisition in a comfortable environment. The equipment used a hair brush-type dry electrode without gel, leading to signal quality comparable to that of a wet electrode [44]. The equipment installation was completed within 5 min. We adopted the dry EEG because it is robust for motion artifacts due to its joint fastening mechanical structure, which enables the frame to adjust to the shape and size of the subject's head. We used 19 electrodes located according to the international 10–20 system of electrode placement: Fp1, Fp2, Fz, F3, F4, F7, F8, Cz, C3, C4, T3, T4, T5, T6, Pz, P3, P4, O1, and O2. The ground channel was at Fpz, and the reference channel was at Pz. We measured EEG signals with a sampling rate of 300 Hz.

Hand pressing force was also measured by a laboratory-made system. The system consisted of eight piezoelectric sensors (Model 208M192, PCB Piezoelectric, Inc., New York, NY, USA) inside an aluminum frame for the fingertips of both hands, except the thumbs. The positions of the sensors were adjusted for the shape of each subject's hand. The forearms were supported by a soft buffer and were fixed to it to steady the arms of each subject. The analog pressure signals obtained from the sensors were then digitized and recorded using LabVIEW.

### 2.3. Experimental Paradigm

The whole experiment consisted of 78 trials. Half of the trials were categorized as the performance of a single task (motor task without mental arithmetic), and the rest were categorized as dual tasks (a motor task with mental arithmetic). Single and dual tasks were then subcategorized as left, right, or both hands, depending on which hands would be used to perform each task. Figure 1 illustrates the whole experimental procedure. Before the experiment, we measured the subject's pre-bimanual maximum voluntary force (pre-BMVF) during the setup of the EEG device. During the pre-BMVF measurement, the subject repeatedly pressed both hands simultaneously three times for 5 s, including one practice trial. The BMVF value was used to set the amplitude of target force, depending on the subject's ability of force production. There were two monitors to display the instruction and motor task feedback.

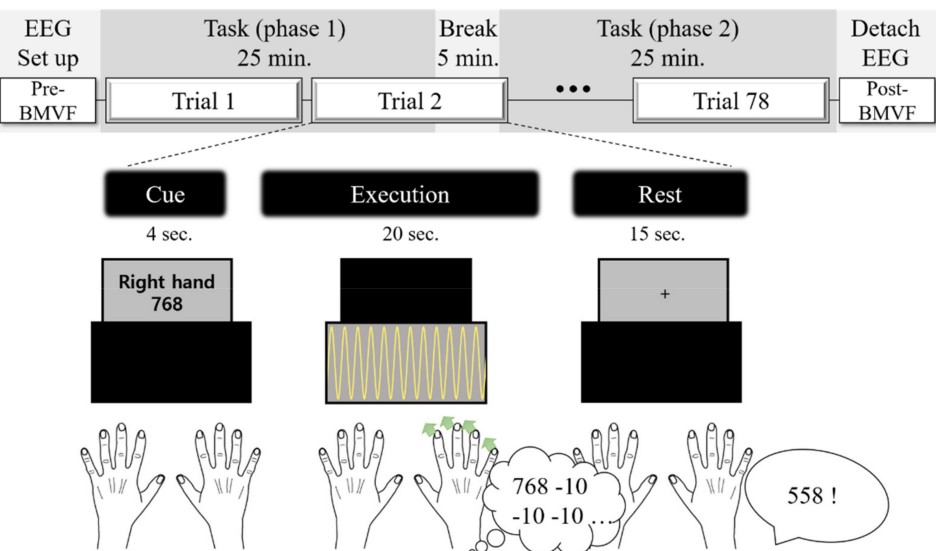

**Figure 1.** Schematic configuration of the experimental procedures. The top flow chart presents the timetable of the experiment from EEG setup to detachment of the device. It takes approximately 2 min to measure the BMVF before and after the task. Each trial consists of a cue (4 s), execution (20 s), and rest (15 s) periods.

Each trial consisted of cue, execution, and rest periods. In the cue period, the upper monitor showed which hand to use (e.g., "right hand") in a single task for 4 s. In the dual-task, the monitor showed an arbitrary three-digit number (ex. "768") and which hand to use (e.g., "right hand") simultaneously for 4 s. In the execution period, the upper monitor was turned off and the lower one was turned on to show the target force line and the force-feedback line. The execution period was 20 s. In the single task, the subject conducted an isometric force control by pressing their fingertips with their left and/or right hand(s). The target force was shown in the monitor by a yellow line shaped as a sine wave with a frequency of 1 Hz, an amplitude of 5% BMVF, and an average of 15% BMVF. The subject was instructed to fit the white line (force production) to the yellow line (target force) by evenly changing the pressure of the fingertips. In the dual-task, the same motor task was performed simultaneously with the mental arithmetic task. For the mental arithmetic task, subjects subtracted 10 from a given three-digit number sequentially. We chose the number 10 as the subtracting number because elderly participants cannot perform complicated mathematics. We tested the subtraction task in our pilot study, which revealed that the subtracting number should be 10 for elderly people to perform the subtraction task properly. Subtraction was done only in the mind. There were six tasks in total and three motor conditions for each single and dual-task. The six tasks were referred to as {Single$^{both}$, Single$^{right}$, Single$^{left}$, Dual$^{both}$, Dual$^{right}$, and Dual$^{left}$}. After conducting the task, the subject rested for 14 s with no conscious thoughts or additional motions, except for giving the final answer of the mental arithmetic in the dual-task. The final answer to the mental calculation was recorded manually to measure the calculation speed, maintain the subjects' concentration, and provide motivation for the subjects.

### 2.4. Preprocessing

The measured EEG signals were down-sampled from 300 Hz to 250 Hz to reduce memory usage. The continuous signals were divided into five frequency ranges: delta (1–4 Hz), theta (4–8 Hz), alpha (8–12 Hz), beta (12–40 Hz), and gamma (40–80 Hz). They were segmented from 0 s to 20 s beginning at the start point of the task execution. The baseline was corrected using the signals for a time interval between −3 s and 0 s.

### 2.5. Estimation of Low- and High-Order Functional Connectivity Values

Low- and high-order functional connectivity (LoFC/HiFC) values were estimated from the preprocessed segments at the five frequency bands. We chose correlation as a measure of FC rather than using other analytic techniques such as coherence, phase-locking value, and phase lag index [45]. Such methods can measure the EEG connectivity while considering nonlinearity and are less sensitive to volume conduction than the correlation method [46]. However, they have a higher computational complexity and are less straightforward compared with the correlation method. The correlation method could be better for BCI applications because it is easy to implement and interpret. By estimating both LoFC and HiFC simultaneously using the multivariate normal distribution (MVND) assumption, we can eliminate the transient correlation between time series by artifacts and provide some nonlinearity in the measurement of connectivity.

Figure 2 illustrates the steps of the procedure for multi-order FC estimation, from top to bottom. Each rectangle represents a data matrix. ch and SR represent the number of electrodes and sampling rate, respectively. The rectangles with the wave and plaid patterns represent the EEG time series and the correlation matrix, respectively. FC estimation consists of two steps designed for the cropped strategy [40] and the MVND [47], both of which require segmentation. In the first step, a trial is segmented by a sliding window having a length of 5 s, with 4.5 s overlapping the neighboring window. This 5 s segment is called a 'crop'. The number of crops is 31 per trial. For the second step, the crop is segmented again into sections to collect the matrix-variate distribution. The time window had a length of 1 s and slid for 0.2 s, resulting in 21 sections per crop.

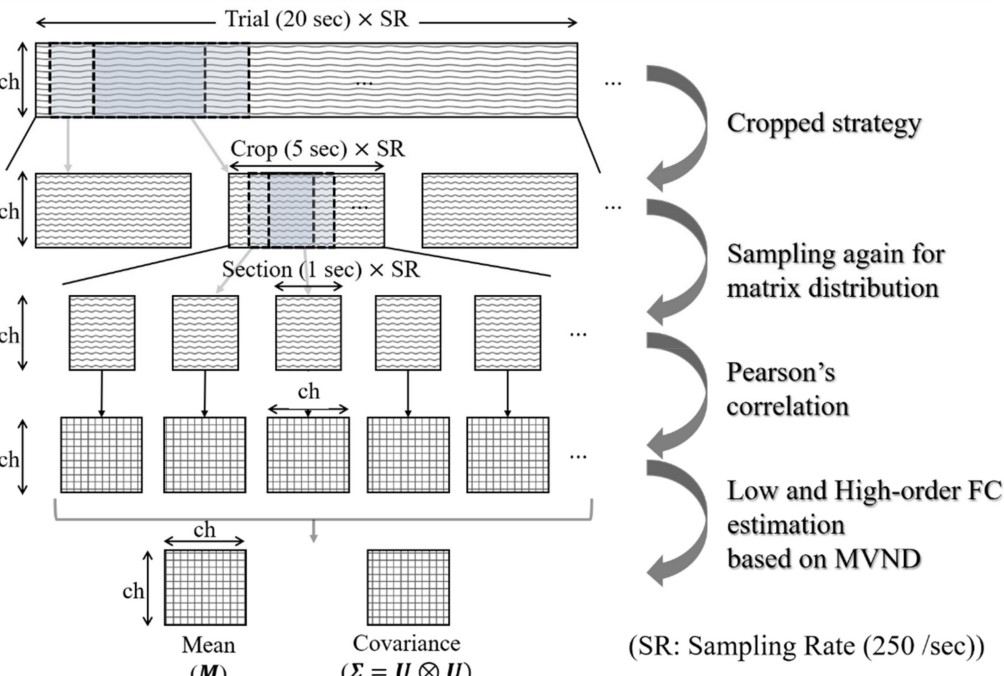

**Figure 2.** Schematic flow chart of the estimation for the low- and high-order functional connectivities from EEG signal segments. The procedure proceeds from top to bottom. Rectangles with wave and plaid patterns represent segments of the EEG time series and correlation matrix, respectively. In the first and second steps, the dashed rectangles inside the rectangles represent sliding windows that form the segments for the next step.

A correlation matrix $r$ was then calculated using the 1 s EEG signals in a section. The $(i, j)$th component of $r$ is $r_{ij}$, which represents the Pearson correlation coefficient between time series $x_i$ and $x_j$. $x_i$ and $x_j$ represent the segmented EEG time series in a *section* at the $i$th and $j$th electrodes, respectively. As a result, 21 correlation matrices were given per

*crop*. We assume that the correlation matrices follow the MVND as $r \sim N(M, U \otimes V)$ and $M \in R^{ch \times ch}$, where $M$, $U$, and $V$ represent the mean and variance among rows, and variance among columns, respectively [48]. The covariance $\Sigma \in R^{ch^2 \times ch^2}$ of the matrix-variate distribution can be described as $U \otimes V$, where $\otimes$ denotes the Kronecker product. We can assume that $U = V$ because $r$ is symmetric. $M$ and $U$, instead of $M$ and $\Sigma$, are used to represent LoFC and HiFC, respectively, because the size of $\Sigma$ is too large [47]. $M$ and $\Sigma$ are calculated based on a maximum likelihood estimation [49,50]. Therefore, we obtained ten FC matrices in a one time-crop because LoFC and HiFC were calculated in five frequency ranges.

### 2.6. Explainable CNN

Figure 3 presents the architecture of the CNN for the classification of a three-dimensional input consisting of low- and high-order FCs at five frequency bands. The cuboids and small spheres represent the input/output feature map and the data points in each stage, respectively. The rectangles with solid lines that cover the data points show the convolutional kernel inside the feature map. The dashed squares cover the pooling range. The whole procedure was divided into two phases. The first phase begins with the input layer. The size of the input variable is [number of electrodes $\times$ number of electrodes $\times$ 10], which has a depth of 10, consisting of [LoFC$_\delta$, HiFC$_\delta$, LoFC$_\theta$, HiFC$_\theta$, LoFC$_\alpha$... HiFC$_\gamma$]. To avoid confusion with the EEG channel, rather than using the term channel, we used the term depth, which is normally used in image processing studies. The subscript characters in LoFC and HiFC represent frequency bands. The first convolution is conducted using 10 2D convolutional filters with a size of $18 \times 1$. The filter length was calculated as [number of electrodes $-$ 1], outputting 10 feature maps with a size of $2 \times 19$. The exponential linear unit (ELU) was applied to the feature map, since ELU outperforms the rectified linear unit in deep learning for EEG signals [40]. Then, 10 2D convolutional filters with a size of $1 \times 18$ were applied in the second convolutional stage. ELU was also applied for nonlinearity. A max pooling layer was applied with a pooling size of $2 \times 2$ to reduce dimensions. The length of the convolutional kernel was defined as [number of electrodes $-$ 1] because this number yielded the best performance in our preliminary study. It was better for the filter's length to be less than the number of electrodes, considering the regional similarity of the FC distribution. However, if the length was too short, the number of layers increased. The more the number of layers, the more the computational cost and time.

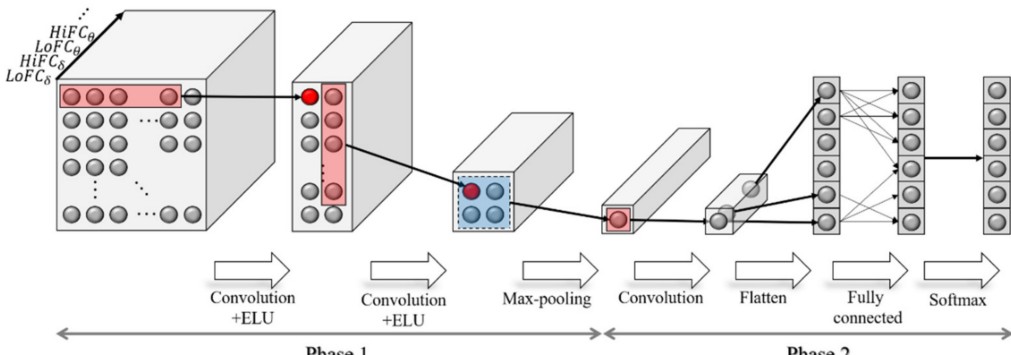

**Figure 3.** Structure of the CNN for the classification of a three-dimensional input consisting of low- and high-order FCs at five frequency bands. The cuboids represent the input and output variables of each process. Circles inside the rectangles represent data points. The first input variable consists of 10 layers of feature maps. Rectangles with solid and dashed lines inside the cuboids represent the convolutional and pooling filters, respectively.

In the second phase, six pointwise convolutional filters were applied to the output of the previous phase [51]. Pointwise convolution applies a $1 \times 1$ sized kernel to a depth direction. The pointwise convolution can combine information of the feature maps [41]. The

final pointwise convolution combined filtered FC information from different feature maps. A linear unit was applied as an activation function, followed by a pointwise convolutional unit. The output of the pointwise convolutional layer was then flattened, fully connected, and passed to a softmax function. This fully connected layer was accompanied by a dropout method to prevent overfitting [52]. The probability of the dropout was 0.5. The softmax function converts the classification output to a probability value, which indicates the probability that the input variable belongs to each class. The class of the input variable can be estimated by finding the maximum value of the output of the softmax function. In ten-fold cross-validation of each subject, the data were randomly shuffled before dividing into folds, and then the average of the performances of the ten-folds was taken. Eight folds were used as a training set, and the rest were used for validation and test sets. The validation set was used to evaluate the performance of the model in the unseen data set. The network was trained with stochastic gradient descent with a batch size of 64 and a constant momentum of 0.9 for 300 epochs. The learning rate was 0.005.

After the training was completed, LRP decomposed the classification output in terms of the relevance values, which represent the contribution of input data by backpropagation [53]. The relevance value $r_i^l$ of the $i$th neuron at layer $l$ was calculated using a local redistribution rule:

$$r_i^l = \sum_j \frac{z_{i \to j}}{\sum_k z_{k \to j}} r_j^{l+1} \ when \ z_{i \to j} = x_i^l w_{i \to j}^{l \to l+1} \tag{1}$$

where $x_i^l$ and $w_{i \to j}^{l \to l+1}$ represent the input of the $i$th neuron at the $l$th layer and the weight connecting the $i^{\text{th}}$ neuron at the $l$th layer to the $j$th neuron at the $(l+1)$th layer, respectively. $z$ is the output of the $i$th neuron. The total amount of relevance in a layer is conserved [53].

### 2.7. Filter Bank Common Spatial Pattern

The filter bank common spatial pattern (FBCSP) was adopted for comparison with the performance of CNN classification using FC. FBCSP exhibits the best performance in decoding the oscillatory activity from EEG signals in the BCI competition [54]. The FBCSP algorithm computes the spatial filter to extract discriminatory information from two classes in a supervised manner. We applied the FBCSP as described in J. Persson et al.'s work [55], after bandpass filtering EEG signals into delta (1–4 Hz), theta (4–8 Hz), alpha (8–12 Hz), beta (12–40 Hz), and gamma (40–80 Hz) ranges. The FBCSP can conduct a form of binary classification. The six-class problem was solved by dividing it into a classifier for the two-class problem, $C_{2-class}$, dealing with ($Dual^{both \cup right \cup left}$, $Single^{both \cup right \cup left}$) and the three-class problem, $C_{3-class}$ ($[Dual \cup Single]^{both}$, $[Dual \cup Single]^{right}$, $[Dual \cup Single]^{left}$), using multilabel transformation in FBCSP. $C_{2-class}$ solved the binary classification problem using FBCSP features with a naive Bayes classifier. $C_{3-class}$ used the combination of three different classifiers for a multiclass problem using a one-vs-one voting strategy [56]. In a testing set, the trained $C_{2-class}$ and $C_{3-class}$ estimated the cognitive label (dual or single) and hand label (both hands or right hand or left hand) of an unseen sample, respectively. Then, the results were combined to predict the label of the input variable among six classes.

### 2.8. Analysis of LRP-Derived Relevance Value by Brain Hemispheres and Regions

We investigated the age difference in LoFC and HiFC within and across the left and right brain hemispheres, and within and across regions, depending on the task and the frequency bands. First, we investigated the age-related overactivation within and across the left and right hemispheres. Functional connectivity between the electrodes located in the same hemisphere (electrodes in the left hemisphere: Fp1, Fz, F3, F7, Cz, C3, T3, Pz, T5, P3, O1, and electrodes in the right hemisphere: Fp2, Fz, F4, F8, Cz, C4, T4, Pz, T6, T4, P2) is called "intra-hemispheric FC". The relevance value of the connectivity between the electrodes in the left and right hemispheres represents an "inter-hemispheric FC" activation. Second, we divided the brain regions to examine the effect of age on the topology of connectivity activation. EEG channels were divided into seven brain areas:

PF (Fp1, Fp2), F (F7, F3, Fz, F4, F8), C (C3, Cz, C4), LT (T3, T5), RT (T4, T6), P (P3, Pz, P4), and O (O1, O2). After the EEG channels were divided into hemispheres or regions, the estimated functional connectivity derived from relevance at each electrode was averaged under each condition.

When investigating inter-regional connectivity, we used graph theory to quantify node connections. We compared the inter-regional connectivity between the younger and older groups using graph theory. In graph theory, networks can be represented as a graph consisting of nodes (points) connected by edges (lines) [57]. In our study, nodes and edges represent the brain regions and age-related different FCs (two-sample *t*-test, $p < 0.05$), respectively. Based on graph theory, we calculated the degree of nodes and the average of all the degrees of the nodes to calculate the numerical representation of the age-dependent changes in inter-regional networks. The degree of nodes represents the number of edges that are incident on the node. The average degree of a network is large when the network is highly connected. The degree of the *i*th node ($i = 1, 2, \ldots, N$) was calculated by:

$$k_i = \sum_{j=1}^{N} S_{ij} \tag{2}$$

where *S* is the adjacency matrix containing binary values for network connectivity, and *N* is the number of brain regions, which was set to seven. Then, the average degree *K* of the network is given by:

$$K = \langle k_i \rangle = \frac{1}{N} \sum_{i=1}^{N} k_i, \tag{3}$$

where the notation $\langle x \rangle$ indicates the mean value of *x*.

## 3. Results

### 3.1. Behavior Results

We obtained behavior performances related to hand force control and mental arithmetic during the six cognitive-motor tasks in the younger and elderly groups. The root-mean square error (RMSE) of hand force was calculated by averaging the squared deviations of hand force relative to the target force for each subject, and it was calculated from the measured hand force between 5 s and 19 s after starting the task to eliminate task-independent variations. The RMSE indicates how well the hand force was accurately produced by the subject. The performance of mental calculations was evaluated by the subtraction speed computed by $\frac{[Target\ number - Answered\ number]}{10}$, where the Target number is the number given on the screen at the beginning of each task. The Answered number is the number received from the subject at the end of each task.

Table 1 presents the RMSE and subtraction speed in the younger and elderly groups. The values are presented as the mean value $\pm$ standard error (SE). We found that only the RMSE, not the subtraction speed, changed according to the age in all tasks. RMSE was significantly larger in the elderly subjects than in the younger subjects (two-sample *t*-test, $p < 0.05$) in all tasks. However, there was no difference between the young adult and elderly groups in the performance of mental arithmetic (two-sample *t*-test, $p > 0.05$).



**Table 1.** RMSE of the Force Control and Speed of Mental Arithmetic Tasks.

| Age | Task | RMSE (kg·m/s$^2$) | | Calculation Speed (Subtraction per Trial) | |
| --- | --- | --- | --- | --- | --- |
| | | Value | Average | Value | Average |
| Young | Single$^{both}$ | 0.18 ± 0.02 | | - | - |
| | Single$^{right}$ | 0.20 ± 0.02 | 0.19 ± 0.02 | | |
| | Single$^{left}$ | 0.20 ± 0.01 | | | |
| | Dual$^{both}$ | 0.19 ± 0.02 | | 20.0 ± 1.24 | |
| | Dual$^{right}$ | 0.20 ± 0.02 | 0.21 ± 0.02 | 18.5 ± 1.13 | 19.4 ± 1.15 |
| | Dual$^{left}$ | 0.23 ± 0.02 | | 19.8 ± 0.98 | |
| Old | Single$^{both}$ | 0.57 ± 0.07 | | | |
| | Single$^{right}$ | 0.55 ± 0.07 | 0.57 ± 0.08 | | |
| | Single$^{left}$ | 0.60 ± 0.11 | | | |
| | Dual$^{both}$ | 0.59 ± 0.09 | | 16.8 ± 1.28 | |
| | Dual$^{right}$ | 0.57 ± 0.09 | 0.60 ± 0.08 | 15.4 ± 1.31 | 15.9 ± 1.30 |
| | Dual$^{left}$ | 0.63 ± 0.08 | | 16.6 ± 1.33 | |

Root-mean squared error (RMSE) and calculation speed are presented as the Mean ± SE. The Average column presents the mean RMSE or calculation speed over three motor conditions for the single and dual tasks.

### 3.2. Classification Accuracy: CNN Results

We classified the combination of LoFC and HiFC into six cognitive-motor tasks for each subject included in the young and elderly groups using CNN. The six tasks were divided into three single and three dual-tasks. Each single or dual-task consisted of three motor conditions.

Figure 4a,b visualize the classification outputs of the test set in the younger and older adults, respectively. The diagonal values represent the ratio of the correctly classified test samples to all the test samples. The off-diagonal values represent the rates of the incorrectly classified samples. The asterisks represent the scores, which show the significant difference calculated by the two-sample *t*-test between age groups in a six-by-six confusion matrix (two-sample *t*-test, $p < 0.05$). In the multi-class problem, recall and precision were calculated by $\left[\sum_{i=1}^{N}\{(Y_i \cap p_i)/Y_i\}\right]/N$ and $\left[\sum_{i=1}^{N}\{(Y_i \cap p_i)/p_i\}\right]/N$, where $N$, $Y_i$ and $p_i$ represent the number of samples, the truth label assigned to the $i$th sample, and the predicted label of the $i$th sample, respectively. The overall accuracy (OA) was obtained by the ratio of the number of correctly classified samples to the number of total samples.

CNN classified the multi-order FC input more accurately in the elderly group than in the young adult group. Especially in the Single$^{both}$, Single$^{left}$, and Dual$^{both}$ tasks, the correctly classified rates were higher in the elderly group than in the young adult group (two-sample *t*-test, $p < 0.05$). The OA was 75.3% in the elderly group, which is 4.6% higher than the accuracy of 70.7% in the young adult group (two-sample *t*-test, $p < 0.05$). We also calculated the Cohen's kappa coefficient to obtain unbiased classification accuracy in the multi-class problem. The kappa value is an evaluation metric for the multi-class classification problem, in which the chance level depends on the number of classes [58]. A kappa coefficient of 0.65 ± 0.034 in the young adult group was lower than that in the elderly group (0.70 ± 0.031) (two-sample *t*-test, $p < 0.05$) for the six-class classification. Classification results are considered to be at an acceptable level when Cohen's kappa value is larger than 0.61, regardless of the number of classes [59].

We verified that the CNN using multi-order FC yielded higher classification accuracy on the six-class classification problem than the FBCSP. The FBCSP yielded significantly lower classification accuracy than CNN using multi-order FC in young adults (two-sample *t*-test, $p < 0.05$, CNN: 70.7%, FBCSP: 42.3%) and elderly (two-sample *t*-test, $p < 0.05$, CNN: 75.3%, FBCSP: 40.0%). Classification accuracy was increased using the proposed method by 67.1% in younger adults and 88.3% in the elderly, compared to the accuracy obtained by using the FBCSP.

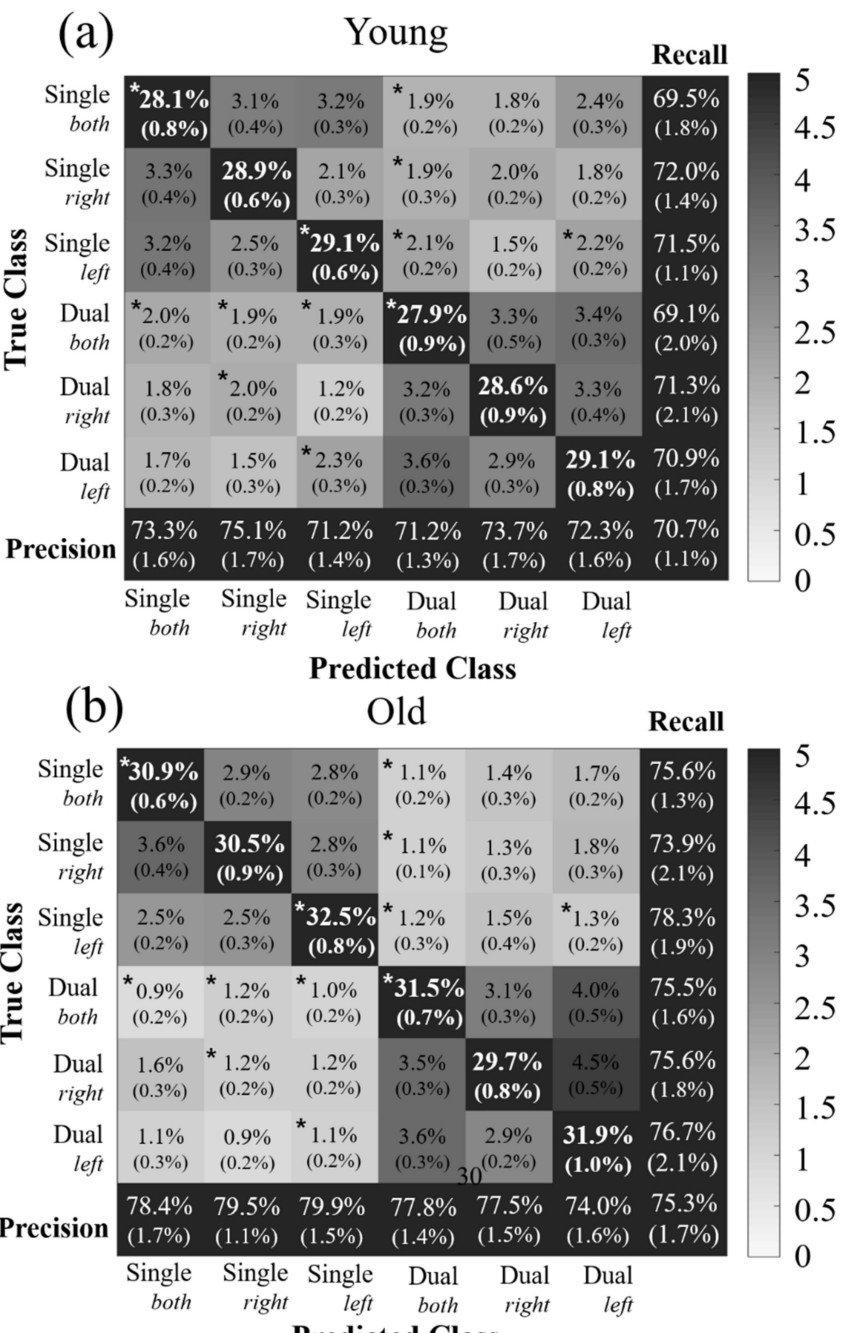

**Figure 4.** Confusion matrix of the six-class classification results using CNN in (**a**) the younger and (**b**) older groups. Each row and column presents the true and predicted class, respectively. The values inside the cells represent the mean and standard error of the intrasubject scores of classification in the corresponding group in the form of Mean (SE). The right-most column and bottom row represent the recall and precision of the classification results. The overall accuracy of each group is presented at the bottom-right corner. The asterisks show the significant difference calculated by the two-sample *t*-test in the classification results between the younger and older groups (two-sample *t*-test, $p < 0.05$), except for the recall and precision lines. The colors of each cell visualize the mean value (%) written inside the cells. Since the maximum value of the off-diagonal values is 5%, the range of the color bar is from 0 to 5 (%). The colors of the cells visualize their off-diagonal values. The asterisks represent the scores, which show the significant difference calculated by the two-sample *t*-test between age groups in a six-by-six confusion matrix (two-sample *t*-test, $p < 0.05$).

### 3.3. Relationship between Classification Accuracies and Behavior Performances

We calculated the relationship between the behavior performance and classification accuracy in the young adult and elderly groups. Cohen's kappa coefficient was used to represent the classification performance for the evaluation of the behavior-accuracy relationship. Pearson's correlation coefficient was calculated between the RMSE and classification accuracy in the single and dual tasks, and between the calculation speed and classification accuracy in the dual-task. For the single task, the RMSE was not correlated with the classification accuracy in the younger (Pearson's correlation, $r = -0.00$, $p > 0.05$) and older (Pearson's correlation, $r = -0.27$, $p > 0.05$) populations. The dual-task also exhibited no correlation between the RMSE and classification accuracy in the younger (Pearson's correlation, $r = 0.01$, $p > 0.05$) and older (Pearson's correlation, $r = -0.10$, $p > 0.05$) groups. A positive correlation between the speed of mental subtraction and the kappa coefficient was found in the elderly group (Pearson's correlation, $r = 0.60$, $p < 0.05$) but not in the younger adults (Pearson's correlation, $r = -0.12$, $p > 0.05$).

### 3.4. LRP Results

We found that the difference in LoFC and HiFC between young adults and elderly group within and across brain regions was depended on the task and frequency bands. We found that the LRP-derived relevance values were higher in the elderly group than in the young adult group, which reflects the age-related FC overactivation. When the relevance was averaged over all the tasks and brain regions, the relevance of the elderly value was greater than that of the young adults by 1.26 times in $LoFC_\beta$ (two-sample $t$-test, $p < 0.05$) and 1.44 times in $HiFC_\beta$ (two-sample $t$-test, $p < 0.05$). In $HiFC_\gamma$, the relevance of the elderly group was 1.23 times greater than that of the young adult group (two-sample $t$-test, $p < 0.05$).

### 3.5. Comparison of LRP-Derived Relevance in Intra- and Inter-Hemispheric FC between Age Groups

We investigated the age-related overactivation for intra- and inter-hemispheric FC. Figure 5 presents the bar graphs for the LRP-derived relevance values of intra- and inter-hemispheric FC in LoFC and HiFC for all tasks in the five frequency bands. The asterisks indicate the significant difference in the relevance values between the younger and elderly groups (two-sample $t$-test, $p < 0.05$). Intra-hemispheric FC exhibited a higher age-related overactivation than inter-hemispheric FC. The overactivation of the intra-hemispheric LoFC for the elderly group was observed in a frequency range above the theta range, mostly in the left hemisphere. In the frequency ranges above beta, the intra-hemispheric network in LoFC was increased in both hemispheres for elderly group. HiFC showed age-related overactivation in the beta and gamma bands. Within similar age groups, there was no difference in intra-hemispheric FC between the left and right hemispheres (Wilcoxon's signed-rank test, $p > 0.05$). However, the intra-hemispheric FC was higher than the inter-hemispheric FC in both groups (Wilcoxon's signed-rank test, $p < 0.05$).

We also investigated the task dependency of the age-related overactivation in intra- and inter-hemispheric connectivities. Figure 6 presents the ratio of FC relevance between the older group and the young adult group. The asterisks indicate that the relevance of FC in the elderly was significantly higher than that in the young adults (two-sample $t$-test, $p < 0.05$). Activations of LoFC and HiFC were higher in the elderly group than in the young adult group for the $Single^{both}$, $Single^{left}$, and $Dual^{both}$ tasks (two-sample $t$-test, $p < 0.05$). For LoFC, the young adult group exhibits the inversion of the relevance ratio, with overactivation for the $Dual^{right}$ and $Dual^{left}$ tasks, but it can be regarded as a meaningless result without statistical significance because the relevance ratio is less than 1.

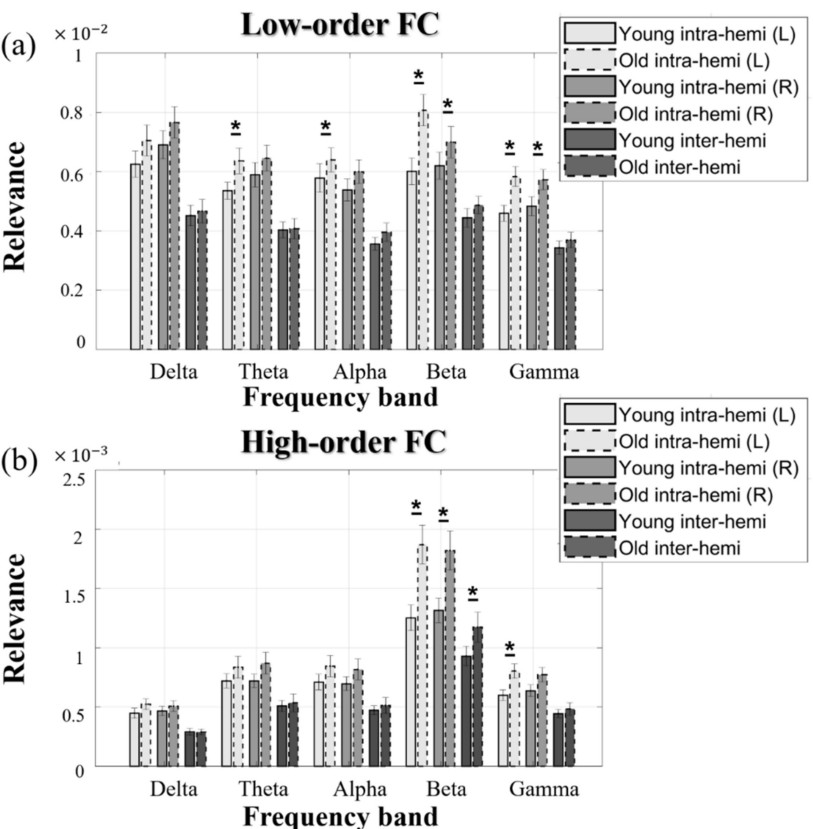

**Figure 5.** Bar graphs of the relevance values for the intra- and inter-hemispheric FC in (**a**) LoFC and (**b**) HiFC. Bars with solid and dashed rectangles represent the younger and older groups, respectively. The gray, medium, and dark gray bars represent the intra-hemispheric FC in the left hemisphere, intra-hemispheric FC in the right hemisphere, and inter-hemispheric FC, respectively. The asterisks indicate the significant differences in the relevance values between the younger and older groups (two-sample *t*-test, $p < 0.05$).

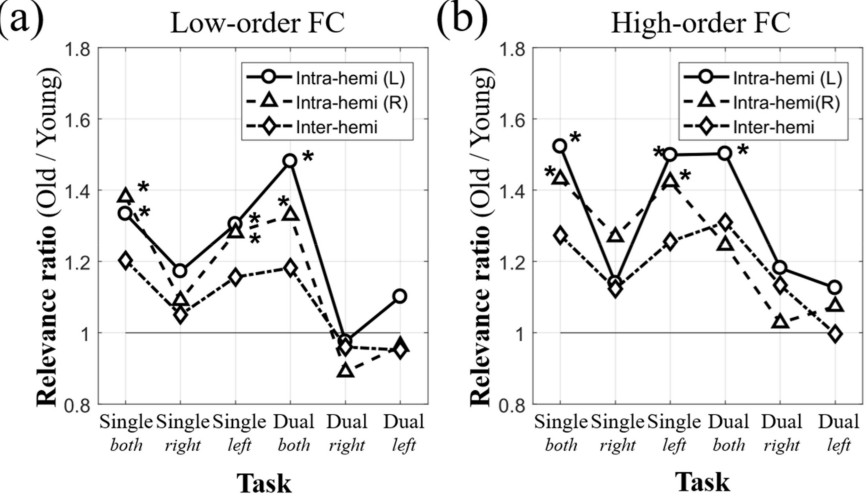

**Figure 6.** Ratios of the relevance values between the elderly and younger adult groups for (**a**) LoFC and (**b**) HiFC. Relevance values were averaged over all brain regions and frequency ranges. Solid lines marked by circles and dashed lines marked by triangles represent the relevance ratios of intra-hemispheric FC in the left and right hemispheres, respectively. The relevance ratios in inter-hemispheric connectivity are indicated by the dash-dotted line with diamond marks. The asterisks show the significant difference in relevance ratios between the younger and older groups (two-sample *t*-test, $p < 0.05$).

### 3.6. Comparison of LRP-Derived Relevance in Intra- and Inter-Regional FCs between Age Groups

The regional contributions of FC to the six cognitive-motor tasks were compared between younger and older groups. Figures 7 and 8 present the bar graphs of the intra-regional relevance of LoFC and HiFC, respectively. Gray and black bars represent the younger and older groups, respectively. The asterisks represent the difference in the intra-regional relevance between the younger and older groups (two-sample $t$-test, $p < 0.05$). Task-specific overactivation occurred in more regions in the single tasks than in the dual tasks for both LoFC and HiFC. In the dual tasks, age-related overactivation in LoFC and HiFC was observed when both hands were used. Among the regions, the prefrontal (PFC) cortex exhibits the largest age-related difference in intra-regional FC activation in LoFC and HiFC. Regarding the degree of overactivation in PFC, LoFC related more to the single task while HiFC related more to bimanual coordination. Table 2 presents the degree of node ($k_i$) in the seven brain regions and the average nodal degree ($K$) of the changed network associated with age in the six tasks. $k_i$ represents the number of edges connecting the brain region (node) with other regions. The relevance results of the inter-regional network show both similarities and differences with those of the intra-regional network. Both show age-related overactivation at the frontal area in the Single$^{both}$, Single$^{left}$, and Dual$^{both}$ tasks. However, compared to the intra-regional FC, inter-regional FC was higher in HiFC than in LoFC, on average. Overall, the average $K$ for all tasks is higher in HiFC (1.26) than in LoFC (0.73).

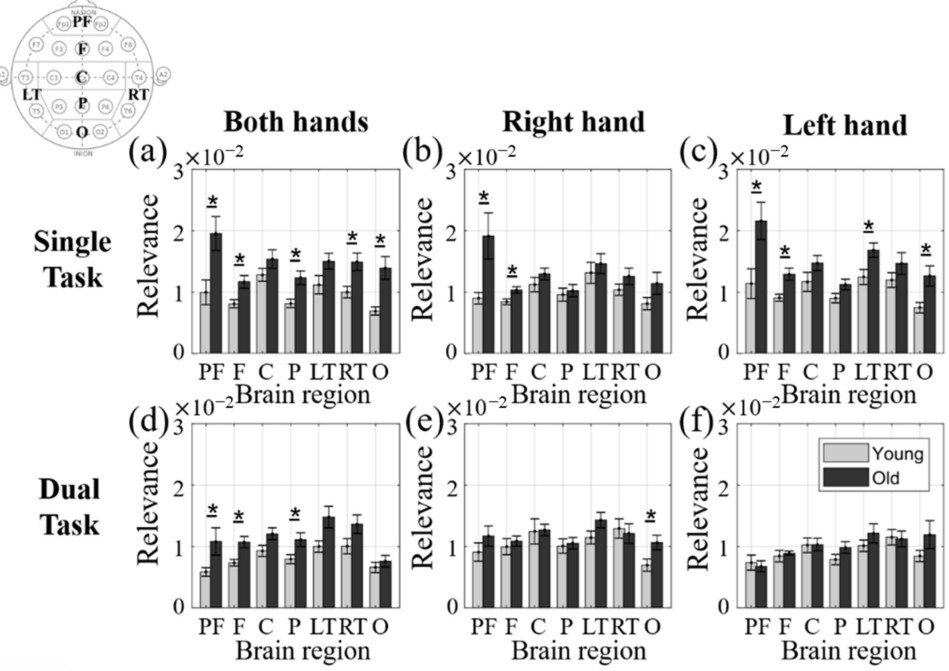

**Figure 7.** Bar graphs of the relevance values of inter-regional LoFC in (**a**) Single$^{both}$, (**b**) Single$^{right}$, (**c**) Single$^{left}$, (**d**) Dual$^{both}$, (**e**) Dual$^{right}$, and (**f**) Dual$^{left}$. The defined brain regions are presented in the top left corner. Gray- and black-colored bars represent the younger and older groups, respectively. The asterisks indicate the significant differences in the relevance values between the younger and older groups (two-sample $t$-test, $p < 0.05$).

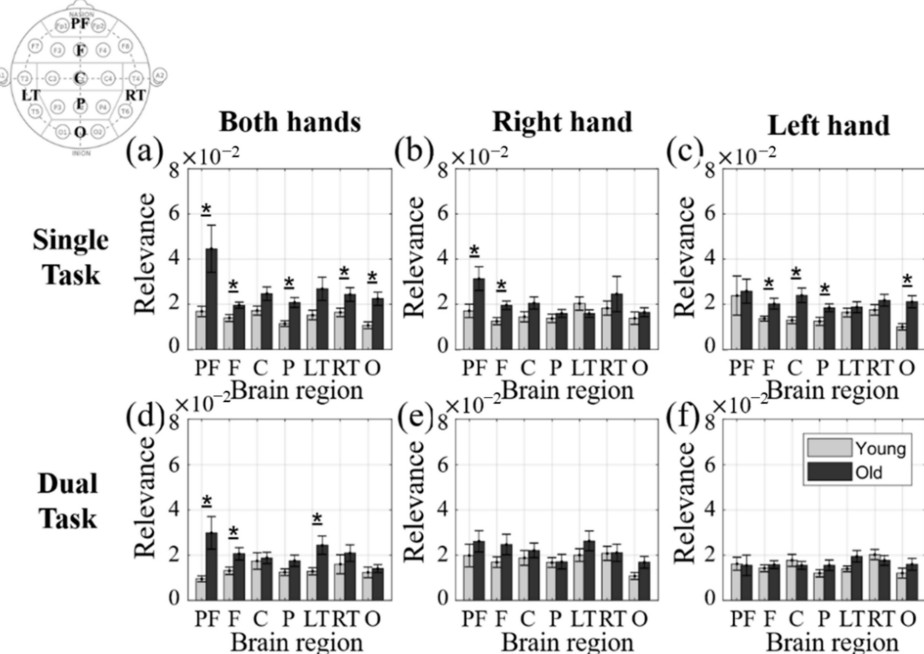

**Figure 8.** Bar graphs of the relevance values of inter-regional HiFC in (**a**) Single$^{both}$, (**b**) Single$^{right}$, (**c**) Single$^{left}$, (**d**) Dual$^{both}$, (**e**) Dual$^{right}$, and (**f**) Dual$^{left}$. The defined brain regions are presented in the top left corner. Gray- and black-colored bars represent the younger and older groups, respectively. The asterisks indicate the significant differences in the relevance values between the younger and older groups (two-sample *t*-test, $p < 0.05$).

**Table 2.** Degree of Node ($k_i$) and Average Nodal Degree ($K$) in the Network of the Inter-regional LoFC and HiFC for Seven Brain Regions.

| | LoFC | Single Both | Single Right | Single Left | Dual Both | Dual Right | Dual Left |
|---|---|---|---|---|---|---|---|
| | PF | 3 | 2 | 2 | 2 | 0 | 0 |
| | F | 2 | 1 | 1 | 3 | 0 | 0 |
| | C | 1 | 0 | 0 | 2 | 0 | 0 |
| $k_i$ | P | 2 | 2 | 0 | 1 | 0 | 0 |
| | LT | 2 | 1 | 1 | 1 | 0 | 0 |
| | RT | 0 | 0 | 0 | 1 | 0 | 0 |
| | O | 0 | 0 | 0 | 1 | 0 | 0 |
| $K$ | | 1.43 | 0.86 | 0.57 | 1.57 | 0 | 0 |
| | **HiFC** | **Single Both** | **Single Right** | **Single Left** | **Dual Both** | **Dual Right** | **Dual Left** |
| | PF | 5 | 3 | 4 | 2 | 0 | 0 |
| | F | 3 | 1 | 2 | 2 | 0 | 0 |
| | C | 2 | 1 | 3 | 1 | 0 | 0 |
| $k_i$ | P | 3 | 0 | 4 | 1 | 0 | 1 |
| | LT | 2 | 0 | 1 | 4 | 0 | 0 |
| | RT | 0 | 0 | 1 | 0 | 0 | 0 |
| | O | 2 | 1 | 3 | 0 | 0 | 0 |
| $K$ | | 2.43 | 0.86 | 2.57 | 1.43 | 0 | 0.29 |

The seven brain regions are the prefrontal (PF), frontal (F), central (C), parietal (P), left-temporal (LT), right-temporal (RT), and occipital (O) areas. $k_i$ and $K$ are calculated by (2) and (3), respectively.

## 4. Discussion

### 4.1. Higher Classification Accuracy in the Elderly Group Than in the Young Adult Group

CNN using FC yielded better performance in the classification of the six-class cognitive-motor tasks than the FBCSP. There are two possible explanations for the improvements with our proposed method. First, the proposed method exploits the fact that the communication of neuronal populations represent high-level cognition, such as attention, emotion, memory, and planning, which is not explained by regional activation. The FBCSP is based on functional segregation of the brain, which is insufficient to understand the neurophysiological source of the human mind in complex tasks. No matter how complex the structure of functional segregation is, it is insufficient to explain how all the brain functions are formed [60]. The use of the state-of-the-art nonlinear classifier, CNN, is the second contributor of substantial accuracy, even when there is a lack of prior knowledge about the physiological basis of FC. Potentially necessary information is often excluded in linear classifiers because it needs to reduce the dimension of the input feature set [41].

We show that the proposed classification method using LoFC and HiFC with CNN is more efficient in multi-class classification for the elderly than for younger adults. The elderly group had a higher classification accuracy, which was revealed in the classification accuracy (elderly: 75.3%, young: 70.7%), confusion matrix (Figure 4), and Cohen's kappa coefficient (elderly: 0.70, young: 0.65) for the six cognitive-motor tasks. This result is not in agreement with the classification result of a recent EEG-BCI study, wherein the elderly group yielded lower classification accuracy than the young adult group [23]. They have average classification accuracies of 66.4% and 82.3% in the older and young adult groups, respectively, for binary classification using EEG oscillatory power with a linear discriminant analysis and common spatial pattern (CSP) [61–63].

There are two reasons for the improvement of the classification for the elderly population in our study. First, FC exhibits larger age-related changes compared to the oscillatory power of EEG [28]. In the brains of the elderly, FC activation contributes more to cognitive functions than regional neural activity. Second, CNN extracts task-related information better than CSP [64] by accounting for age-related changes in brain activation. The FC pattern of an aging brain contains different temporal and spatial properties compared with that of a younger brain. While CSP omits some meaningful information during feature selection, CNN can use as much information as possible for machine learning, regardless of age-related changes in the FC patterns. The results imply that a combination of FC with CNN would be preferable to a spectral feature with a linear classifier, especially for BCI systems that target older people.

Our results also support CRUNCH by finding that the behavioral performance in the mental calculation is positively correlated with the classification accuracy in the elderly, but not with that in the young adult group. The connectivity of the aging brain works hard to produce the best performance, which explains the similar performances of mental arithmetic between the younger and older groups (Table 1). The age-related overactivation may not be reflected in the performances of the motor tasks due to physical impairment [24].

### 4.2. Age-Related Compensatory Overactivation in the Prefrontal Cortex

LRP provides specific evidence and characteristics for the age-related compensatory activities in LoFC and HiFC. After CNN learning through hidden layers provides a descriptive explanation of the activity, the LRP-derived relevance values enable us to describe processes and causes of that activity through backpropagation. Our results indicate that the age-related overactivation of FC was dominant at the PFC (Figures 7 and 8, Table 2). This finding is consistent with those of previous works on age-related compensation in the brain network [65,66]. The frontal overactivation of FC is found while performing a single task, which implies that the decline in motor function in an aged person is because of physical impairment rather than a deficit in compensatory neural correlates [67]. Dual tasks evoke less age-related overactivation than single tasks because the increasing difficulty by cognitive load decreases the age-related overactivation [24]. PFC overactivation is evoked

by bimanual coordination, even in a dual-task, because PFC is associated with motor coordination [68]. The frontal lobe is recruited more during bimanual movement in the elderly than in their younger counterparts [69]. The LRP results explain why the proposed method exhibited higher classification performance in the elderly group than in the young adult group, despite neurocognitive decline from aging.

*4.3. Age-Related Increase in Functional Connectivity within Hemispheres Rather Than across Hemispheres*

In this study, FC was increased in the elderly group compared to the young adult group within hemispheres rather than across hemispheres (Figure 5). There is evidence supporting the hemispheric cooperation model in which inter-hemispheric interaction increases to compensate for the neural deficit of one hemisphere in the aging brain [70]. However, our results show intra-hemispheric overactivation rather than inter-hemispheric overactivation against the perspective of hemispheric cooperation. This is because age-related hemispheric cooperation occurs in tasks where the activation of the younger brain is lateralized [71]. In this study, LoFC and HiFC were evenly activated for the left and right hemispheres in younger adults (Figure 5). Therefore, the aging brain does not need to increase long-distance interaction across hemispheres.

*4.4. Compensatory Overactivity in Higher-Order FC*

Although the effects of aging on neural networks between brain regions have been studied previously, the age-related changes in the relationship between the networks have rarely been studied. In this study, we found that normal aging not only affects the connectivity between brain regions (LoFC) but also the connectivity between brain networks (HiFC). LoFC and HiFC have similarities and differences in task-specific patterns and age-related compensation. Both inter-regional LoFC and HiFC increase with age at the frontal area for the same tasks (Figures 7 and 8). This suggests that LoFC and HiFC have similar functional roles triggered in a specific location. They may cooperate to complement performance for tasks hindered by age-related functional declines.

On the other hand, HiFC has distinctive characteristics compared to LoFC. Age-related activities in HiFC were mostly observed in the beta band (Figure 5). Beta activity is responsible for high-level cognitive functions [72] such as the integration of sensory inputs for older people [73]. Therefore, we can assume that HiFC is associated with higher-level cognitive processes such as functional integration. Moreover, HiFC shows higher inter-regional overactivation than LoFC (Table 2). Considering that LoFC serves to gather information and that HiFC plays a role in abstracting that information [74], the results indicate that HiFC contributes to the preserved or improved function of high-level cognitive processing in the elderly by integrating neural inputs from different areas. The inter-regional increase of HiFC in the aging brain could explain why some elderly people can successfully achieve complicated tasks despite their neurocognitive decline [67].

## 5. Conclusions

The primary finding of this study is that age-related compensatory overactivation in multi-order FC results in higher accuracy in multi-class BCIs for the elderly than for the young adult population. We designed and used a CNN to maximize the ability to extract information from high-dimensional FC maps in both age groups. The proposed method improved the classification accuracy in six-class problems by 67.1% in the young adult group and 88.3% in the elderly group compared to FBCSP. Classification accuracy was 75.3% for the elderly group, which was 4.6% higher than that of the young adult group. LRP, one of the explanatory techniques of deep learning, gave a neurophysiological explanation of the impact of age-related changes in the FCs on classification performance in the younger and older populations. LoFC and HiFC in the prefrontal cortex were more activated in the aging brain compared to the younger brain, depending on the type of task. High-order FC increased in the beta band to integrate neural inputs from different brain regions in the aging brain. Therefore, our results provide ways and reasons for a

multi-order FC with explainable CNN to be an optimal method for BCI applications in the elderly. Future BCI research for older people should further investigate the impact of age by including higher orders of functional connectivity to develop appropriate features, with consideration for age-related neurophysiological changes. This study can potentially extend the use of BCIs to healthy elderly people to improve their quality of life.

**Author Contributions:** Conceptualization, S.D.; methodology, S.D.; software, S.D. and Y.J.; valida-tion, B.Y., S.D. and Y.J.; formal analysis, S.D.; investigation, S.B. and B.Y.; resources, S.D. and B.Y.; data curation, S.D.; writing—original draft preparation, S.D., Y.J., S.B. and B.Y.; writing—review and editing, J.J.; visualization, S.D.; supervision, J.J.; project administration, J.J.; funding acquisition, J.J. All authors have read and agreed to the published version of the manuscript.

**Funding:** This work was supported by Institute of Information & communications Technology Planning & Evaluation (IITP) grant funded by the Korean government (MSIT) (No. 2017-0-00451; Development of BCI based Brain and Cognitive Computing Technology for Recognizing User's Intention using Deep Learning).

**Institutional Review Board Statement:** This study was approved by the Institutional Review Board of the Korea University (NO. 17-126-A-2). Also, this experiment was conducted according to the guidelines of the Declaration of Helsinki.

**Informed Consent Statement:** Informed consent was obtained from all subjects involved in the study. Written informed consent has been obtained from the participants to publish this paper.

**Data Availability Statement:** Data sharing is not applicable to this article.

**Conflicts of Interest:** The authors declare no conflict of interest.

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
