# Peer review of "Explainable Convolutional Neural Network to Investigate Age-Related Changes in Multi-Order Functional Connectivity"

_electronics, doi:10.3390/electronics10233020_

Round 1

Reviewer 1 Report

The title of the paper contains "Explainable CNN", which is the reason why I have been interested in reviewing the paper. My main criticism regarding the paper is related to this part, because I feel that the paper does not contribute to the research domain of Explainable AI (XAI) in general nor to explainable CNNs in special. I did not rate the Novelty since I am not an expert in BCIs and FC.

The term "explainable" in conjunction with AI (or CNN) is a hot research topic in computer science, because CNNs produce classifiers that cannot be explained by default. Researchers are trying hard to compute an explanation based on the learned CNN. But I saw no mention of this research area. Hence, I assume that the paper title is not appropriate and you mean something else.

In line 523 you state that CRUNCH is supported by the "classification accuracy". So you draw conclusions about "the brain", based on the accuracy of a CNN? This is not sound. So far nobody can explain what the CNN really learned. Somebody else could have used Random Forests or LSTM or what so ever AI, get a different accuracy and draw different conclusions. In general terms, it is not sound to reason about the real world based on classification results of some AI. The other way round is ok: You can reason about an AI based on objective knowledge of the real world.

The merit of the paper may be improved accuracy due to the usage of CNNs. I am not an expert in BCIs. Thus, I cannot rate whether that is sufficient for publication. But the "explainable" part of the paper is not acceptable.

Otherwise, the paper is written very well and the text is supported by many pictures. However, I have some remarks:

Line 261: "After learning with forward propagation". What exactly does that mean? Based on this information, it would be impossible to reproduce what you have done.

Figure 4: What do the colors mean? There is a scale from 0 ... 5, but no explanation is given in the figure caption.

Line 485: This is text from the template and must be removed.

Author Response

I would like to resubmit the revised paper entitled, “Explainable Convolutional Neural Network to Investigate the Age-Related Changes in Multi-Order Functional Connectivity,” to Special issue of Advanced Technologies and Challenges in Brain Machine Interface of Electronics. Our manuscript undergwent extensive English revisions by Wordvice. We marked revisions to the manuscript using “Track Changes” function of MS where and what changes have been made.

We’d like to thank you for your helpful comments and corrections. Our paper was revised according to reviewer’s comments and editorial corrections as following:

REVIEWER REPORT(S):

Reviewer: 1

Comments and suggestions:

The title of the paper contains "Explainable CNN", which is the reason why I have been interested in reviewing the paper. My main criticism regarding the paper is related to this part, because I feel that the paper does not contribute to the research domain of Explainable AI (XAI) in general nor to explainable CNNs in special. I did not rate the Novelty since I am not an expert in BCIs and FC.

The term "explainable" in conjunction with AI (or CNN) is a hot research topic in computer science, because CNNs produce classifiers that cannot be explained by default. Researchers are trying hard to compute an explanation based on the learned CNN. But I saw no mention of this research area. Hence, I assume that the paper title is not appropriate and you mean something else.

In line 523 you state that CRUNCH is supported by the "classification accuracy". So you draw conclusions about "the brain", based on the accuracy of a CNN? This is not sound. So far nobody can explain what the CNN really learned. Somebody else could have used Random Forests or LSTM or what so ever AI, get a different accuracy and draw different conclusions. In general terms, it is not sound to reason about the real world based on classification results of some AI. The other way round is ok: You can reason about an AI based on objective knowledge of the real world.

The merit of the paper may be improved accuracy due to the usage of CNNs. I am not an expert in BCIs. Thus, I cannot rate whether that is sufficient for publication. But the "explainable" part of the paper is not acceptable.

  • We used the term “explainable” because we adopted the LRP method to explain the effect of the brain signal (functional connectivity), which is the input, on the classification result of CNN. We used the term "explainable" because we were able to reveal the learning layer of the CNN network's input layer through back-propagation using the LRP method. The reason AI is considered a "black box" is that when AI is trained on data, the final output is known and its accuracy is high, but the weights of the previous layers are not known. LRP can provide the influence of the input layer on the final output through the process of reducing the ‘cause’ obtained through validity propagation to weights and dissecting it. We are eligible to use the term “explainable”.

Otherwise, the paper is written very well and the text is supported by many pictures. However, I have some remarks:

Line 261: "After learning with forward propagation". What exactly does that mean? Based on this information, it would be impossible to reproduce what you have done.

  • We modified “After learning by forward propagation” to “After training was finished” to avoid confusion. Forward propagation refers to the calculation and storage of intermediate variables (including outputs) for a neural network in the order from the input layer to the output layer.

Figure 4: What do the colors mean? There is a scale from 0 ... 5, but no explanation is given in the figure caption.

  • We added “The colors of each cell visualize the mean value (%) written inside the cells. Because the maximum value of off-diagonal values is 5 %, the range of the color bar is 0 to 5. The colors of the cells visualize the off-diagonal values.” in the caption of Figure 4 to avoid confusion.

Line 485: This is text from the template and must be removed.

  • We removed the paragraph from the template.

Reviewer 2 Report

Comments and suggestions:

1. Line 77: More citations could be added for FC applications along with description of more network parameters that have been used such as global efficiency, degree of connectivity and characterstic path lengths.

Examples induce:

Passive BCI's using functional connectivity for emotion recognition with various network parameters:

[1] Tiwari, Abhishek, and Tiago H. Falk. "Fusion of Motif-and spectrum-related features for improved EEG-based emotion recognition." Computational intelligence and neuroscience 2019 (2019).

[2] Gupta, Rishabh, and Tiago H. Falk. "Relevance vector classifier decision fusion and EEG graph-theoretic features for automatic affective state characterization." Neurocomputing 174 (2016): 875-884.

for emotion recognition as well as:

[1] Sengupta, Anwesha, et al. "Analysis of loss of alertness due to cognitive fatigue using motif synchronization of eeg records." 2016 38th Annual International Conference of the IEEE Engineering in Medicine and Biology Society (EMBC). IEEE, 2016.

for alertness detection.

2. Line 77-78: "However, feature extraction requires prior knowledge of the input data and often removes relevant information." This is a big claim and would be better to cite

3. Line 80: "..and is hard to interpret because it addresses properties of the network, not the network itself.." This line doesn't make a lot of sense, as characterizing network properties is equivalent to characterizing a network itself.

4. Line 85-89: The font size is not consistent and keeps on changing.

5. Line 96: What are RCs? I think its a typo and should be FC?

6. Line 256-257: Three questions emerge from the statement: 1) Does the 10 fold cross validaiton shuffle the data randomly before dividing into folds or the data is divided with no temporal shuffling of the epochs? 2) Most of the hyper-parameters have already been set, in such a case, what is the use of the validation set? Is a 10 fold cross validation performed or performance is only evaluated on one of the 10 folds for each subject?

7. Line 260: The line mentions the "initial" learning rate. Has the rate been updated in any way with epoch training? if not, why call it the initial rate?

8. Table 1: cntHb is not defined anywhere

9: Lines 340-350: They are completely out of place and describe the wrong results from the table. They seem to be about fNIRS which is not mentioned anywhere in the protocol before. This needs to be rectified.

10. Line 366: How was the significant difference calculated? Is it done accuracy results for each subject or total accuracy of each of the folds?

11. Line 402: While a correlation with mental arithmetic task was found, I think a backward subtraction of 10 is very intuitive and easy to perform. Why was 10 chosen as the subtracting number? As it could easily  be done in bigger steps thus corrupting the results. 

12. Figure 6: For the dual right and left task we see an inversion of the ratio with overactivation for the younger group for the lower FC. How do the authors explain this observation?

13. Line 485-487: These lines again feel out of place and probably got overlooked and should be deleted.

14. Line 538: This is a major claim and would be helped by some supporting references.

Author Response

I would like to resubmit the revised paper entitled, “Explainable Convolutional Neural Network to Investigate the Age-Related Changes in Multi-Order Functional Connectivity,” to Special issue of Advanced Technologies and Challenges in Brain Machine Interface of Electronics. Our manuscript undergwent extensive English revisions by Wordvice. We marked revisions to the manuscript using “Track Changes” function of MS where and what changes have been made.

We’d like to thank you for your helpful comments and corrections. Our paper was revised according to reviewer’s comments and editorial corrections as following:

Reviewer: 2

Comments and suggestions:

  1. Line 77: More citations could be added for FC applications along with description of more network parameters that have been used such as global efficiency, degree of connectivity and characterstic path lengths.

Examples induce:

Passive BCI's using functional connectivity for emotion recognition with various network parameters:

[1] Tiwari, Abhishek, and Tiago H. Falk. "Fusion of Motif-and spectrum-related features for improved EEG-based emotion recognition." Computational intelligence and neuroscience 2019 (2019).

[2] Gupta, Rishabh, and Tiago H. Falk. "Relevance vector classifier decision fusion and EEG graph-theoretic features for automatic affective state characterization." Neurocomputing 174 (2016): 875-884.

for emotion recognition as well as:

[1] Sengupta, Anwesha, et al. "Analysis of loss of alertness due to cognitive fatigue using motif synchronization of eeg records." 2016 38th Annual International Conference of the IEEE Engineering in Medicine and Biology Society (EMBC). IEEE, 2016.

for alertness detection.

  • We added more references with the statement of “Several studies of passive BCI were tried to use FC for emotion recognition [70, 71] and alertness detection [72] with various network parameters such as global efficiency, degree of connectivity and characteristic path lengths.”:
    • [70] Tiwari, A., & Falk, T. H. (2019). Fusion of Motif-and spectrum-related features for improved EEG-based emotion recognition. Computational intelligence and neuroscience, 2019.
    • [71] Gupta, R., & Falk, T. H. (2016). Relevance vector classifier decision fusion and EEG graph-theoretic features for automatic affective state characterization. Neurocomputing, 174, 875-884.
    • [72] Sengupta, A., Tiwari, A., Chaudhuri, A., & Routray, A. (2016, August). Analysis of loss of alertness due to cognitive fatigue using motif synchronization of eeg records. In 2016 38th Annual International Conference of the IEEE Engineering in Medicine and Biology Society (EMBC) (pp. 1652-1655). IEEE.
  1. Line 77-78: "However, feature extraction requires prior knowledge of the input data and often removes relevant information." This is a big claim and would be better to cite
  • We added a reference [73] to support our claim of “However, feature extraction requires prior knowledge of the input data and often removes relevant information [73].”
    • [73] Chu, C., Hsu, A. L., Chou, K. H., Bandettini, P., Lin, C., & Alzheimer's Disease Neuroimaging Initiative. (2012). Does feature selection improve classification accuracy? Impact of sample size and feature selection on classification using anatomical magnetic resonance images. Neuroimage, 60(1), 59-70.
  1. Line 80: "..and is hard to interpret because it addresses properties of the network, not the network itself.." This line doesn't make a lot of sense, as characterizing network properties is equivalent to characterizing a network itself.
  • We changed a sentence "..and is hard to interpret because it addresses properties of the network, not the network itself." to “..and is hard to interpret intuitively” to avoid a confusion.
  1. Line 85-89: The font size is not consistent and keeps on changing.
  • We fixed the font size by 10 through the Line 85-89.
  1. Line 96: What are RCs? I think its a typo and should be FC?
  • We modified “RCs” to “FCs” in Line 96.
  1. Line 256-257: Three questions emerge from the statement: 1) Does the 10 fold cross validaiton shuffle the data randomly before dividing into folds or the data is divided with no temporal shuffling of the epochs? 2) Most of the hyper-parameters have already been set, in such a case, what is the use of the validation set? Is a 10 fold cross validation performed or performance is only evaluated on one of the 10 folds for each subject?
  • 1) Yes, the data is randomly shuffled before dividing into folds.
  • 2) The validation set is used to evaluate the performance of the model in the unseen data set. The test for validation set prevents overfitting of the training.
  • 3) For each subject, 10-fold cross validation is performed and the performance of ten folds are averaged for final results.
  • We added these in the line 256-257.
  1. Line 260: The line mentions the "initial" learning rate. Has the rate been updated in any way with epoch training? if not, why call it the initial rate?
  • We modified “initial learning rate” to “learning rate”.
  1. Table 1: cntHb is not defined anywhere
  • We modified “cntHb” to “Young” in Table 1.

      9: Lines 340-350: They are completely out of place and describe the wrong             results from the table. They seem to be about fNIRS which is not                           mentioned anywhere in the protocol before. This needs to be rectified.

  • We removed the paragraph and rearranged the table.
  1.  Line 366: How was the significant difference calculated? Is it done accuracy results for each subject or total accuracy of each of the folds?
  • The significant difference was calculated using two-sample t-test. Two-sample t-test is also known as the independent samples t-test and used to test whether the unknown population means of two groups are equal or not statistically. The accuracy result is calculated for each subject. We added the test method in Line 366.
  1. Line 402: While a correlation with mental arithmetic task was found, I think a backward subtraction of 10 is very intuitive and easy to perform. Why was 10 chosen as the subtracting number? As it could easily be done in bigger steps thus corrupting the results.
  • We added comments “We chose number ten as the subtracting number because the elderly participants cannot perform complicated mathematics. We tested the subtraction task in our pilot study, revealing that subtracting number should be ten for elderly people to perform subtraction task properly.” in 3. Experimental paradigm. (Line 170)
  1. Figure 6: For the dual right and left task we see an inversion of the ratio with overactivation for the younger group for the lower FC. How do the authors explain this observation?
  • We didn’t interpret an inversion of the ratio for the dual right and left task for younger group for the lower FC, because they didn’t show the statistical significance. Although the mean value of ratio of the relevance values between the elderly and younger adult groups in dual right and left was lower than 1, it is regarded as meaningless results without statistical significance. We added this in line 468.

  1. Line 485-487: These lines again feel out of place and probably got overlooked and should be deleted.
  • We deleted the paragraph.
  1. Line 538: This is a major claim and would be helped by some supporting references.
  • We added a reference [74] to support our claim:
    • [74] King, B. R., Fogel, S. M., Albouy, G., & Doyon, J. (2013). Neural correlates of the age-related changes in motor sequence learning and motor adaptation in older adults. Frontiers in human neuroscience, 7, 142.